# WeaveNet for Approximating Assignment Problems

## Abstract

Assignment, a task to match a limited number of elements, is a fundamental problem in informatics. Many assignment problems have no exact solvers due to their NP-hardness or incomplete input, and their approximation algorithms have been studied for a long time. However, individual practical applications have various objective functions and prior assumptions, which usually differ from academic studies. This gap hinders applying the algorithms to real problems despite their theoretically ensured performance. In contrast, a learning-based method can be a promising solution to fill the gap. To open a new vista for real-world assignment problems, we propose a novel neural network architecture, *WeaveNet*. Its core module, *feature weaving layer*, is stacked to model frequent communication between elements in a parameter-efficient way for solving the combinatorial problem of assignment. To evaluate the model, we approximated one of the most popular non-linear assignment problems, stable matching with two different *strongly NP-hard* settings. The experimental results showed its impressive performance among the learning-based baselines. Furthermore, we achieved better or comparative performance to the state-of-the-art algorithmic method, depending on the size of problem instances.

## 1 Introduction

From multiple object tracking to job matching, assignment problems can represent a wide variety of applications. An assignment problem is typically defined on a bipartite graph, a graph with two sets of nodes $A$ and $B$ with edges $E = A \times B$ ($N = |A|$, $M = |B|$, $N \geq M$). On the graph, the task is to find a matching $m \in \{0, 1\}^{A \times B}$ (a set of edges represented as a binary matrix) that satisfies constraints and/or maximizes objectives. Depending on real-world scenes, there must be various objectives and constraints for $m$. A typical constraint is a one-to-one correspondence (i.e., every node has at most one matched partner in $m$) and, for simplicity, we always assume it in this paper.

Matching stability is another example of such constraints. It is a non-linear constraint first introduced for a hospital-student assignment problem (Gale and Shapley, 1962) based on the preferences of hospitals among students and vice versa. We say a matching $m$ is unstable when there exist $a \in A$ and $b \in B$ which are unmatched in $m$ ($m_{ab} = 0$) but both prefer each other more than their partner in $m$. We can obtain a stable matching $m$ in $O(N^2)$ by the Gale-Shapley (GS) algorithm (Gale and Shapley, 1962). However, when $m$ is expected to have the minimum difference in the total satisfactions between sides $A$ and $B$ (known as sex-equal stable matching), the problem becomes *strongly NP-hard*[1] (Kato, 1993; McDermid and Irving, 2014).

In addition to the NP-hardness, we also face difficulties to obtain the best assignment when assignment candidates may randomly disappear (e.g., multiple object tracking with occlusions (Emami *et al.*, 2020) or joint matching in multi-person pose estimation (Cao *et al.*, 2017)). In such cases, we need

---

[1] *strongly NP-hard* is a subclass of NP-hard and considered more complex than general NP-hard problems

Submitted to 35th Conference on Neural Information Processing Systems (NeurIPS 2021). Do not distribute.

to compensate for the inputs of incomplete information by its stochastic properties. The traditional methods often use sub-optimal approximations to avoid solving complex assignment problems. A differential assignment model can be a future option that enables end-to-end training for such applications.

Toward such future applications, this paper aims to propose an effective and promising differential solver for assignment problems. The contribution of this paper is four-fold:

1. We proposed *WeaveNet*, a novel neural network architecture for assignment problems and *set-encoder*, a novel local structure.
2. We proposed a novel technique, *split batch normalization*, to deal with a strong asymmetry in input distributions for sides $A$ and $B$.
3. We focused on stable matching, a classical non-linear assignment problem actively studied even in recent years, and proposed a novel evaluation protocol[2] with *pseudo costs*, which enables us to compare learning-based solvers and algorithmic solvers directly.
4. We achieved a better performance with the state-of-the-art algorithmic baseline when $N = 20$, and a comparative performance when $N = 30$. We also outperformed any learning-based baselines with a large margin.

## 2 Related work

Despite the recent research interest in deep learning technology, we hardly have a fully differential assignment solver. As long as authors know, there are two past attempts to solve assignment problems by a fully differential model. Li (2019) has tried to solve stable matching by multiple layer perceptrons (MLP). Their contribution is in the proposed relaxation of the non-linear stability constraint to a differential loss function. However, the MLP is too redundant to learn the assignment strategy without overfitting. In addition, the proposed auxiliary loss to maintain the output to be one-to-one matching (symmetric doubly stochastic function) overly constrains the solution search space. In this study, we propose a parameter-efficient differential model and a weaker but sufficient constraint to output a one-to-one matching.

The second attempt is made by Gibbons *et al.* (2019), where Deep Bipartite Matching (DBM) is proposed. They tested their model with the weapon-target assignment (WTA) problem. WTA is a classical NP-hard problem whose state-of-the-art algorithm (Ahuja *et al.*, 2007) could find optimal solution when $N \leq 20$ in the experiment although there is no theoretical guarantee. In this sense, we can consider WTA is empirically easier than sex-equal stable matching, for which we have no such efficient solvers even for $N = 5$. In addition, DBM is trained in a supervised manner or with reinforcement learning, which is hard to apply to a larger $N$. Furthermore, the implementation details are not completely explained, and their dataset and source codes are not publicly available. Finally, the architecture of DBM is still parameter-redundant, and their local structure is sub-optimal. In this study, we propose a more parameter-efficient two-stream architecture, *WeaveNet*, with a novel local structure, *set-encoder*, both of which have significant impacts on the performance.

In addition to the above methods, it is natural to consider using graph convolutional networks (GCNs). However, there are no GCN methods for assignment problems due to the over-smoothing problem (Li *et al.*, 2018; Oono and Suzuki, 2020). Because any graph-convolutional layer summarizes the output with neighboring nodes, its smoothing effect eliminates expressive power for node classification. To avoid such elimination, GIN (Xu *et al.*, 2019), the state-of-the-art GCN method, stacks only two layers for a node classification task. Such elimination is critical for an assignment-problem solver because it needs to identify any slight difference through frequent communication among nodes. Unlike GIN, our model retains edge-wise features rather than node-wise summaries, which does not cause the smoothing problem. Therefore, we can make the model very deep, which any traditional graph convolutional networks cannot.

## 3 Stable matching problem as a benchmark task

To evaluate learning-based assignment solvers, we adopt two *strongly NP-hard* variants of stable matching. They have been actively studied for a long time (Kato, 1993; Iwama *et al.*, 2010; Dworczak,

---

[2]The source code and datasets are included in this submission and will be publicly available.

2016; Gupta *et al.*, 2019) and their state-of-the-art algorithm by Tziavelis *et al.* (2019) must be a
strong baseline against learning-based methods. Hence, we set these two variants as the benchmark
task for learning-based assignment problems.

An instance $I$ of a stable matching problem consists of two sets of agents $A$ and $B$ on a bipartite
graph. Fig. 1 illustrates an example of $I$. Each agent $a_i$ in $A$ ($0 < i \le N$) has a preference list $p_i^A$,
which is an ordered set of elements in $B$ and $p_{ij}^A = rank(b_j; p_i^A)$ is the index of $b_j$ in the list $p_i^A$. $a_i$
prefers $b_j$ to $b_{j'}$ if $p_{ij}^A < p_{ij'}^A$. Similarly, each agent $b_j$ in $B$ ($0 < j \le M$) has a preference list $p_j^B$.

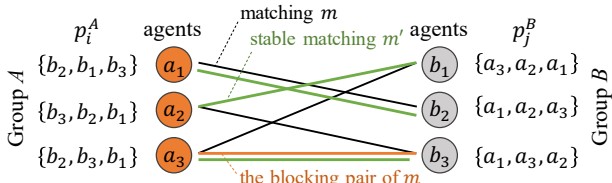

Figure 1: An example of assignment, where $m$ (black edges) is not stable due to the blocking pair
(the orange edge), while $m'$ (green edges) is stable.

For a matching $m$, we say that an unmatched pair $\{a_v, b_w\}$ ($m_{vw} = 0$) blocks $m$ if $a_v$'s partner
$b_j$ ($m_{vj} = 1$) and $b_w$'s partner $a_i$ ($m_{iw} = 1$) satisfy the conditions $p_{vw}^A < p_{vj}^A$ and $p_{wv}^B < p_{wi}^B$. Here,
$\{a_v, b_w\}$ is called a blocking pair (the orange edge blocks a matching of black edges in the figure).

A matching is stable if (and only if) it includes no blocking pair (the green edges in the figure). Note
that $I$ always has at least one stable matching, and the Gale-Shapley (GS) algorithm can find it in
$O(N^2)$. However, the GS algorithm has a biased nature, where one side is prioritized and the other
side only gets the least preferable result among all the possibilities of stable matching.

To compensate for the unfairness, we can introduce diverse objectives to maintain a stable matching
fair. Among them, the following two objectives make the stable matching problem *strongly NP-hard*.
The first one is **Sex equality cost** ($SEq$) (Gusfield and Irving, 1989). It focuses on the unfairness
brought by the gap between the two sides' satisfaction and defined by

$$SEq(m; I) = |P(m; A) - P(m; B)|, \quad P(m; A) = \sum_{\{a_i, b_j\} \in m} p_{ij}^A, \quad P(m; B) = \sum_{\{a_i, b_j\} \in m} p_{ji}^B. \quad (1)$$

The other is **Balance cost** ($Bal$) (Feder, 1995; Gupta *et al.*, 2019), which is a compromise between
side-equality and overall satisfaction. It is defined by

$$Bal(m; I) = \max(P(m; A), P(m; B)). \quad (2)$$

In the proposed evaluation protocol, we minimize either cost while maintaining stable one-to-one
matching.

**Input and output data format for stable matching**   Learning-based approximation is realized by
a trainable function $F$ that outputs a matching $\hat{m} \in [0, 1]^{N \times M}$, which is an $N \times M$ matrix. As
for the input, the value range of the preference rank depends on the problem size, which causes a
range shift of the input distribution. To avoid such shift, we linearly re-scale[3] the rank of preference
$p_{ij}^*$ ($* \in \{A, B\}$) from $[1, N]$ to a normalized score $s_{ij}^*$ ranged in $(0, 1]$ to make it invariant to $N$,
where 1 for the highest rank. Then, we obtain the input as matrices $S^A$ and $S^B$, where $s_{ij}^A$ is the
$ij$-element of $S^A$.

# 4   Deep-learning-based fair stable matching with WeaveNet

## 4.1   WeaveNet

One of the required properties of $F : (S^A, S^B) \to \hat{m}$ is to take all the agents' preference into
account when determining the presence of each edge in the output $\hat{m}$. Li (2019) implemented this by

---

[3]The details of this linear re-scaling are based on Li (2019) and described in A.1. Note that sections numbered
with capital letters appear in the supplementary material.

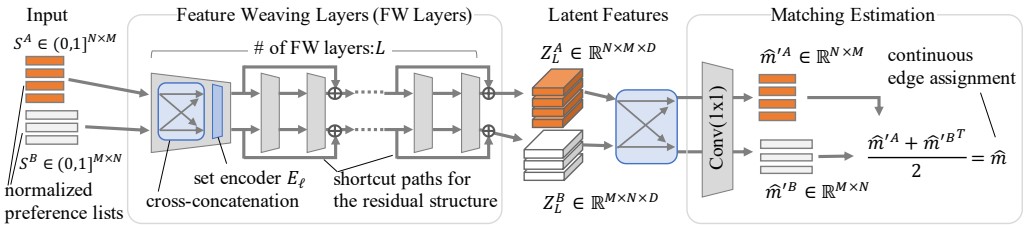

Figure 2: WeaveNet architecture. $L$ feature weaving layers are stacked with shortcut paths to be a deep network. The encoded features are fed into Conv($1 \times 1$) layer to obtain logits ($\hat{m}'^A$, $\hat{m}'^B$). The output $\hat{m}$ will be binarized in prediction phase to represent a matching.

MLPs, where $S^A$ and $S^B$ are destructured and concatenated into a single flat vector (with the length of $2NM$) and fed to the MLP. Its output (a flat vector with the length of $NM$) is restructured into a matrix $\hat{m}$. The MLP model, however, would face difficulties due to the following four problems.

**(a)** Preference lists of multiple agents are encoded by independent parameters, though they share a format so that we could efficiently process them in the same manner.

**(b)** MLP only supports a fixed-size input, so training different models for different cases of $N$ becomes mandatory.

**(c)** $F$ should be permutation invariant, which means the matching result should be unchanged even if we shuffle the order of agents in $S^A$ and $S^B$, but MLP does not satisfy.

**(d)** A shallow MLP model may be insufficient to approximate an exact solver for the NP-hard problem when $N$ is large.

To address the above weaknesses of MLP, we propose the feature weaving network (**WeaveNet**) which has the properties of (a) **shared encoder**, (b) **variable-size input**, (c) **permutation invariance**, and (d) **residual structure**. The WeaveNet, as shown in Fig. 2, consists of $L$ feature weaving (FW) layers. It has two streams of $A$ and $B$. In a symmetric manner, each stream models the agent's act of selecting the one on the opposite side while sharing weights to enhance the parameter efficiency. The shortcut paths at every two FW layers make them residual blocks, which allows the model to be as deep as possible. We explain its details as follows.

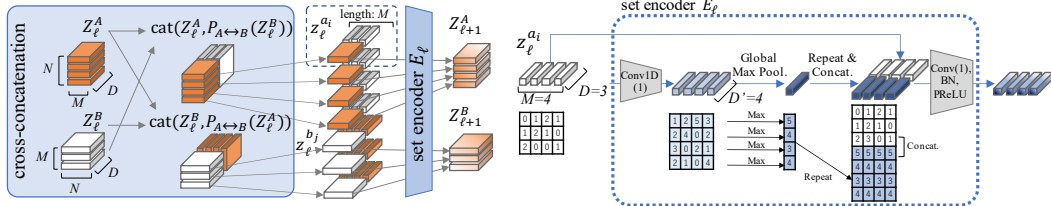

Figure 3: Feature weaving layer orthogonally concatenates the weftwise and warpwize components ($Z_\ell^A$ and $Z_\ell^B$) in a symmetric way (cross-concatenation). Then, the concatenated tensors are separated into $z_\ell^{a_i}$ (or $z_\ell^{b_j}$), which represents a set of outgoing edges from agent $a_i$ (or $b_j$), and independently fed to $E_\ell$.

Figure 4: Illustration of the process in set encoder $E_\ell$, where $z_\ell^{a_i}$ (colored in white) is once encoded to $D'$ channel features (colored in pale blue), then max-pooled to obtain statistics in the feature set (colored in blue). The statistics information is concatenated to each input feature and further encoded (color in a gradation).

Fig. 3 illustrates the detail of a single FW layer, which is the core architecture of the proposed network. FW layer is a two-stream layer whose inputs consist of a *weftwise* component $Z_\ell^A$ and a *warpwise* component $Z_\ell^B$, which are the output of $(l-1)$-th layer and $Z_0^A = S^A$ and $Z_0^B = S^B$ for the first layer. The two components are symmetrically concatenated in each stream (**cross-concatenation**). Then these concatenations are separated into agent-wise features, each of which is a set of outgoing-edge features of an agent (indicating the preference from that agent to every matching candidate). These features are processed by the encoder $E_\ell$ **shared by every agent in both $A$ and $B$**. As for an encoder that can embed **variable-size** input in a **permutation invariant** manner, we adopted the structure inspired by DeepSet (Zaheer *et al.*, 2017) and PointNet (Qi *et al.*, 2017) (Fig. 4), which

consists of two convolutional layers with kernel size 1 and a set-wise max-pooling layer, followed by batch-normalization and PReLU activation. We refer to this structure as *set encoder*.

**Mathematical formulations**   $Z_\ell^A$ in Fig. 3 is a third-order tensor whose dimensions, in sequence, corresponding to the agent, candidate, and feature dimension, with a size of $(N, M, D)$. Similarly, $Z_\ell^B$ has a size of $(M, N, D)$. The **cross-concatenation** is defined as

$$Z_\ell'^A = cat(Z_\ell^A, P_{A \leftrightarrow B}(Z_\ell^B)), \tag{3}$$

where $P_{A \leftrightarrow B}$ swaps the first and second dimensions of the tensor, and $cat(\{Z_1, Z_2, \ldots\})$ concatenates the features of two tensors $Z_1$, $Z_2$. $Z_\ell'^A$ is then sliced into agent-wise features $z_\ell^{a_i}$ and we obtain $Z_{\ell+1}^A = (E_\ell(z_\ell^{a_i})|0 < i \leq N)$, which is also a third-order tensor (and fed to the next layer). We can calculate $Z_{\ell+1}^B$ in a symmetric manner (with the same encoder $E_\ell$).

After the process of $L$ FW layers, $Z_L^A$ and $Z_L^B$ are further cross-concatenated and fed to the matching estimator (in Fig. 2). It outputs a non-deterministic edge assignment $\hat{m}$. In the training phase, $\hat{m}$ is input to an objective function, and the loss is minimized. In the prediction phase, the matching is obtained by binarizing $\hat{m}$. In this sense, matching estimation through a neural network can be considered as an approximation by relaxing the binary assignment space $\{0, 1\}^{N \times M}$ into a continuous assignment space $[0, 1]^{N \times M}$.

**Asymmetric variant with split batch normalization**   WeaveNet is designed to be fully symmetric for $S^A$ and $S^B$. Hence, it satisfies the equation $F(S^A, S^B) = F(S^B, S^A)^\top$. This condition ensures that the model architecture cannot distinguish the two sides $A$ and $B$ innately. This property is beneficial when mathematically fair treatment between $A$ and $B$ is desirable. However, when inputs from $A$ and $B$ are differently biased (e.g., the two sides have different trends of preference or the objective is asymmetric for $A$ and $B$), this symmetric treatment degrades the performance. To eliminate the bias difference without losing the parameter-efficiency, we further propose to **a)** apply batch normalization independently for each stream (*split batch normalization*), and **b)** adding a side-identifiable code (e.g., 1 for $A$ and 0 for $B$) to $Z_0^A$ and $Z_0^B$ as a $(D+1)$-th element of the feature. We call this variant "asymmetric".

## 4.2   Relaxed continuous optimization for fair stable matching

Generally, a combinatorial optimization problem has discrete objective functions and conditions, which are not differentiable. To optimize the model in an end-to-end manner without inaccessible ground truth, we optimize the model by relaxing such discrete loss functions into continuous ones.

Assume we target to obtain a fair stable matching that has the minimum $SEq$, for example. Then, we have the following three loss functions.

$\mathcal{L}_m$ conditions the binarization of $\hat{m}$ to represent a matching.
$\mathcal{L}_s$ conditions the matching to be stable.
$\mathcal{L}_f$ minimizing the fairness cost $SEq$ of the matching

The overall loss function is defined as

$$\mathcal{L}_{\text{fsm}}(\hat{m}) = \lambda_m \mathcal{L}_m + \frac{1}{2} \sum_{m \in \{\hat{m}^A, \hat{m}^B\}} \left(\lambda_s \mathcal{L}_s(m) + \lambda_f \mathcal{L}_f(m)\right), \tag{4}$$

where $\hat{m}^A = \text{softmax}(\hat{m})$ and $\hat{m}^B = \text{softmax}(\hat{m}^\top)$.

An important advantage of learning-based approximation is its flexibility. We can modify the above loss functions to easily obtain other variants. For example, removing $\mathcal{L}_f$ in Eq.(4) leads to standard stable matching, and replacing $\mathcal{L}_f$ with $\mathcal{L}_b$ (which minimizes $Bal$) leads to balanced stable matching, as follows:

$$\mathcal{L}_{\text{sm}}(\hat{m}) = \lambda_m \mathcal{L}_m + \frac{1}{2} \sum_{m \in \{\hat{m}^A, \hat{m}^B\}} \lambda_s \mathcal{L}_s(m), \tag{5}$$

$$\mathcal{L}_{\text{bsm}}(\hat{m}) = \lambda_m \mathcal{L}_m + \frac{1}{2} \sum_{m \in \{\hat{m}^A, \hat{m}^B\}} \left(\lambda_s \mathcal{L}_s(m) + \lambda_b \mathcal{L}_b(m)\right). \tag{6}$$

**One-to-one matching constraint**   $\hat{m}$ can be safely converted into a binarized matching by column-wise or row-wise $\arg\max$ operation when it is a symmetric doubly stochastic matrix (Li, 2019). To satisfy this condition, we defined $\mathcal{L}_m$ with an average of the cosine distance as

$$\mathcal{L}_m(\hat{m}^A, \hat{m}^B) = 1 - \frac{1}{2}(\mathrm{C}(\hat{m}^A, \hat{m}^B) + \mathrm{C}(\hat{m}^B, \hat{m}^A)),$$

$$\mathrm{C}(\hat{m}^A, \hat{m}^B) = \frac{1}{N} \sum_{i=0}^{N} \frac{\hat{m}_{i*}^A \cdot \hat{m}_{*i}^B}{\|\hat{m}_{i*}^A\|_2 \|\hat{m}_{*i}^B\|_2}, \tag{7}$$

where $\hat{m}_{i*}^A$ means the $i$-th row of $\hat{m}^A$. This formulation binds $\hat{m}$ to be a symmetric[4] doubly stochastic matrix when $\mathcal{L}_m(\hat{m}^A, \hat{m}^B) = 0$. The advantage of this implementation against the original one in Li (2019) is described in B.1 with additional experimental results.

**Blocking pair suppression**   As for $L_s$, we used the function proposed in Li (2019), which is

$$\mathcal{L}_s(\hat{m}; I) = \sum_{(v,w) \in A \times B} g(a_v; b_w, \hat{m}) g(b_w; a_v, \hat{m})$$

$$g(a_i; b_w, \hat{m}) = \sum_{b_j \neq b_w} \hat{m}_{ij} \cdot \max(S_{iw}^A - S_{ij}^A, 0)$$

$$g(b_j; a_v, \hat{m}) = \sum_{a_i \neq a_v} \hat{m}_{ji}^\top \cdot \max(S_{jv}^B - S_{ji}^B, 0), \tag{8}$$

where $g(a_i; b_w, \hat{m})$ is a criterion known as ex-ante justified envy, which has a positive value when $a_i$ prefers $b_w$ more than any $b_j$ in $\{b_j | j \neq w, \hat{m}_{ij} > 0\}$. This is the same for $g(b_j; a_v, \hat{m})$. Hence, $\{a_v, b_w\}$ becomes a (soft) blocking pair when both $g(a_v; b_w, \hat{m})$ and $g(b_w; a_v, \hat{m})$ are positive.

**Fairness measurements**   $\mathcal{L}_f, \mathcal{L}_b$ minimize $SEq(m; I), Bal(m; I)$, respectively, and are defined as

$$\mathcal{L}_f(\hat{m}; I) = \frac{1}{N}|S(\hat{m}; A) - S(\hat{m}; B)| \quad \mathcal{L}_b(\hat{m}; I) = -\frac{1}{N}\min(S(\hat{m}; A), S(\hat{m}; B)), \tag{9}$$

where

$$S(\hat{m}; A) = \sum_{i=1}^{N} \sum_{i=j}^{M} \hat{m}_{ij} \cdot S_{ij}^A, \ S(\hat{m}; B) = \sum_{j=1}^{M} \sum_{i=1}^{N} \hat{m}_{ij} \cdot S_{ji}^B. \tag{10}$$

# 5   Experiments

We evaluated WeaveNet with different sizes of $N$. First, with test samples of $N < 10$, we compared its performance with learning-based baselines and optimal solutions obtained by a brute-force search. Second, we compared WeaveNet with algorithmic baselines at $N = 20,\ 30$, where neither existing learning-based methods nor brute-force search work. We also demonstrated the generalization ability of WeaveNet under the mismatched training/test dataset distributions. Third, we demonstrated the performance of WeaveNet at $N = 100$. Note that we always assume $M = N$ hereafter.

**Sample generation protocol**   In the experiments, we used the same method as Tziavelis *et al.* (2019) to generate synthetic datasets that draw preference lists from the following distributions.

**Uniform (U)**   Each agent's preference towards any matching candidate is totally random, defined by a uniform distribution $\mathcal{U}(0, 1)$ (larger value means prior in the preference list).

**Discrete (D)**   Each agent has a preference of $\mathcal{U}(0.5, 1)$ towards a certain group of $\lfloor 0.4N \rfloor$ popular candidates, while $\mathcal{U}(0, 0.5)$ towards the rest.

**Gauss (G)**   Each agent's preference towards $i$-th candidate is defined by a Gaussian distribution $\mathcal{N}(i/N, 0.4)$.

**LibimSeTi (Lib)**   Simulate real rating activity on the online dating service LibimSeTi (Brozovsky and Petricek, 2007) based on the 2D distribution of frequency of each rating pair $(p_{ij}^A,\ p_{ji}^B)$.

---

[4]Here a (possibly non-square) matrix $\hat{m}$ $(N \geq M)$ is symmetric if and only if $\hat{m}_{i*} = \hat{m}_{*i},\ (0 < i \leq M)$.

Choosing the above preference distributions for group $A$ and $B$ respectively, we obtained five different dataset settings, namely UU, DD, GG, UD, and Lib. We randomly generated 1,000 test samples and 1,000 validation samples for each of the five distribution settings.

**Training protocol** We trained any learning-based models 200k total iterations at $N \leq 30$ and 300k at $N = 100$, with a batch size of 8. We randomly generated training samples at each iteration based on the distribution of each dataset and used the Adam optimizer (Kingma and Ba, 2015). We set learning rate 0.0001 and loss weights $\lambda_s = 0.7$, $\lambda_m = 1.0$, $\lambda_f = \lambda_b = 0.01$ based on a preliminary experiment (see A.4).

**Pseudo fairness costs for comparing learning-based results with algorithmic results** Note that for learning-based methods, there is a trade-off between fairness scores and stable matching rate. Hence they may violate the constraints of stable one-to-one matching and yield an $SEq$ or $Bal$ even lower than the ideal value. To compare the methods fairly with traditional algorithmic methods, we evaluate our methods using pseudo $SEq$ ($pSEq$) and pseudo $Bal$ ($pBal$) cost, in which the cost of violation cases is replaced by the worst result of the GS algorithm (prioritizing each side once and adopting the *worse* one).

### 5.1 Comparison with learning-based methods ($N = 3, 5, 7, 9$)

**Baselines and ablations** In this experiment, we show results obtained by following baselines and WeaveNet variants. **MLP** is the model proposed in Li (2019). **GIN** is the state-of-the-art GCN model proposed in Xu *et al.* (2019). We use each (normalized) preference list as a node feature and bipartite edges as the graph structure. After two graph-convolution calculations, as MLP, we destructed the node-wise embeddings and concatenated them into a single vector, which is fed to one Linear layer to output $\hat{m}$. **DBM** is the model in Gibbons *et al.* (2019). **SSWN** is the single-stream WeaveNet, which is equivalent to a DBM adopting the set-encoder of WeaveNet. **WN** is the standard WeaveNet.

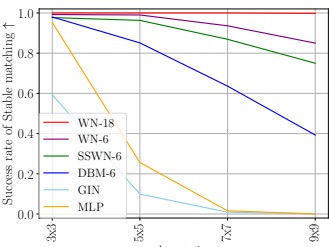 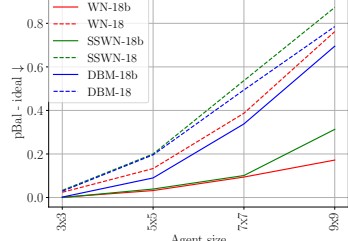

Figure 5: Change of the success rates of stable matching ($\uparrow$) according to $N$

Figure 6: Change of $pSEq-$ ideal scores ($\downarrow$) according to $N$.

Figure 7: Change of $pBal-$ ideal scores ($\downarrow$) according to $N$.

Fig. 5 shows the success rates of finding a stable matching, where we trained models to minimize Eq. (5), considering only the stable matching constraints. Since MLP and GIN have size-dependency, we trained the models independently for $N = 3, 5, 7, 9$. The other models were trained with $N = 10$ and tested on $N = 3, 5, 7, 9$. We maintained models with $L = 6$ layers (the model names are noted as XXX-6) to have a similar number of parameters with MLP for $N = 5$ (see A.5), while WN-18 is prepared to demonstrate the full performance (with the residual blocks).

MLP and GIN can hardly find stable matchings when $N \geq 5$. Note that the number of total cases for size $N$ instances is estimated by $N!^{2(N-1)}$. Hence, when $N = 3$, there are only 1,296 cases at most, and the test set will fully overlap with the training set. In contrast, when $N = 5$, we have $4.3 \times 10^{16}$ cases, and the overlap is negligible. Therefore, we can say that methods working only with $N = 3$, such as MLP and GIN, have little generalization ability.

DBM performs better than MLP but obviously worse than SSWN and WN. The performance gain of SSWN-6 over DBM-6 represents the advantage of the set-encoder. Similarly, the improvement of WN-6 over SSWN-6 shows the benefit of the two-stream architecture. Finally, that of WN-18 over WN-6 demonstrates the impact of stacked layers on the performance. Fig. 15 of the appendix shows

some additional baselines, including a performance of our $L_m$ against the original one proposed in Li (2019).

Figs. 6 and 7 show $pSEq$ and $pBal$ (their difference from the ideal values[5]), respectively. XXX-18f/b are trained to minimize Eqs. (4) and (6), respectively[6]. We omitted MLP and GIN due to their poor performance in Fig. 5. In the results, both SSWN and WN largely outperformed DBM, which again proved the advantage of the set-encoder. WN performed better than SSWN for larger $N$, owing to the parameter efficiency of the two-stream architecture. Note that the performance gain of XXX-18f/b from XXX-18 proved the flexibility of general learning-based methods for customized objective functions.

### 5.2 Comparison with algorithmic methods ($N = 20, 30$)

As the algorithmic methods, we prepared four baselines. **GS** is the *better* result of applying the GS algorithm to prioritize each side once, which runs in $O(N^2)$. **PolyMin** minimizes some alternative fairness costs (the regret and egalitarian costs, which can be solved in $O(N^2)$ and $O(N^3)$, respectively (Gusfield, 1987; Irving *et al.*, 1987; Feder, 1992)). **DACC** by Dworczak (2016) is an approximate algorithm that runs in $O(N^4)$. **PowerBalance** is the state-of-the-art method that runs in $O(N^2)$.

**WN-60f/b(20/30)** is WeaveNet with $L = 60$ layers trained with samples of $N = 20$ and $N = 30$. Note that we used the asymmetric variant for UD and Lib. Moreover, we do not involve any traditional learning-based methods in this part since they scored clear performance drops with increasing $N$ (see Fig. 5) and the problem size of $N = 20, 30$ is clearly beyond their capabilities, but an ablation with WeaveNet variants is reported in B.2.

Table 1: Average $SEq$ ($\downarrow$) and success rate of stable matching ($\uparrow$). Bold and underlined scores shows the **best** and second best ones, respectively.

| Agents ($N \times M$) Datasets (Dist. Type) | 20 × 20 | | | | | 30 × 30 | | | | |
|---|---|---|---|---|---|---|---|---|---|---|
| | UU | DD | GG | UD | Lib | UU | DD | GG | UD | Lib |
| GS | 41.89 | 18.81 | 19.52 | **70.97** | 19.66 | 94.03 | 43.46 | 36.56 | **163.77** | 39.78 |
| PolyMin | 19.93 | 11.83 | 20.57 | 87.08 | 18.47 | 35.52 | 21.21 | 37.37 | 209.62 | 31.85 |
| DACC | 24.34 | 20.13 | 23.07 | 101.75 | 20.40 | 40.87 | 34.35 | 40.59 | 240.48 | 33.88 |
| Power Balance | 16.28 | 8.93 | 17.07 | 71.09 | 15.40 | 18.45 | 11.05 | **27.22** | 163.90 | 21.57 |
| WN-60f(20) ($pSEq$) | 12.23 | **6.37** | **15.50** | 71.31 | 14.59 | 25.21 | 11.38 | 29.36 | 172.63 | 23.53 |
| Stably Matched (%) | 98.90 | 99.50 | 99.40 | 99.60 | 99.30 | 94.60 | 97.30 | 95.70 | 91.30 | 97.70 |
| WN-60f(30) ($pSEq$) | 12.16 | 6.53 | 15.56 | 71.34 | **14.53** | **18.30** | **10.52** | 27.39 | 170.35 | 22.17 |
| Stably Matched (%) | 99.10 | 99.40 | 99.40 | 99.50 | 99.80 | 98.10 | 99.00 | 98.00 | 93.90 | 98.60 |

Table 2: Average $Bal$ ($\downarrow$) and success rate of stable matching ($\uparrow$).

| Agents ($N \times M$) Datasets (Dist. Type) | 20 × 20 | | | | | 30 × 30 | | | | |
|---|---|---|---|---|---|---|---|---|---|---|
| | UU | DD | GG | UD | Lib | UU | DD | GG | UD | Lib |
| GS | 89.14 | 146.16 | 108.36 | **140.53** | 68.62 | 184.05 | 322.05 | 225.49 | **312.12** | 137.59 |
| PolyMin | 74.19 | 140.99 | 108.04 | 145.28 | 66.94 | 144.48 | 306.28 | 224.13 | 324.54 | 130.79 |
| DACC | 78.49 | 146.71 | 110.06 | 151.34 | 68.75 | 150.71 | 316.18 | 227.52 | 337.43 | 133.59 |
| Power Balance | 73.28 | 140.12 | 106.92 | 140.55 | 65.89 | **138.04** | 302.30 | **220.26** | **312.12** | **126.96** |
| WN-60b(20) ($pBal$) | **71.89** | 138.79 | **106.20** | 140.84 | 65.85 | 141.49 | 302.73 | 221.92 | 317.60 | 130.58 |
| Stably Matched (%) | 98.50 | 98.80 | 99.50 | 99.70 | 98.80 | 96.10 | 96.70 | 95.00 | 88.90 | 93.80 |
| WN-60b(30) ($pBal$) | 72.33 | **138.75** | 106.65 | 140.79 | **65.84** | 140.40 | **301.59** | 223.02 | 313.59 | 127.93 |
| Stably Matched (%) | 98.00 | 99.10 | 98.60 | 99.80 | 99.10 | 97.00 | 98.60 | 93.70 | 98.80 | 98.00 |

We show the results in Tables 1 and 2. When $N = 20$, except for UD, the proposed method constantly performed better than any algorithmic methods for both $SEq$ and $Bal$. When $N = 30$, they are comparative. For UD, GS performed even better than PowerBalance. That means that the ideal solution constantly prioritizes one side (a kind of the strongest bias). Since we designed the WeaveNet architecture to treat the sides evenly, this is the most challenging situation for WeaveNet. Nonetheless,

---

[5]1.362, 2.534, 3.746, 4.694 in $SEq$ and 2.406, 6.478, 11.956, 18.706 in $Bal$ for $N = 3, 5, 7, 9$.

[6]We early-stopped the training for DBM-18f/b at 80k due to a sudden overfit after the epoch.

the proposed split batch normalization (with the side-identifiable code) achieved similar performance to GS and PowerBalance. We show the performance drop with the fully symmetric version in B.2 of the appendix, which is also interesting from the ethical viewpoint. It is noteworthy that the model trained with $N = 20$ performs well even with $N = 30$, which indicates that the method has generalizability for size difference.

**Generalization ability for different distributions**    A learning-based method should have a certain generalizability for input distribution shifts. To test the ability, we evaluated the performance of models trained with UU, DD, and GG on test sets of different distributions.

Table 3: The generalizability of WeaveNet (trained/tested with $N = 30$).

| WN-60f | | test | | | |
|---|---|---|---|---|---|
| train | | UU | DD | GG | Avg. |
| UU | $pSEq$ | 18.30 | 25.81 | 29.09 | 21.10 |
| | Stably Matched (%) | 98.10 | 94.90 | 93.60 | 95.53 |
| DD | $pSEq$ | 171.27 | 10.52 | 77.36 | 86.38 |
| | Stably Matched (%) | 2.80 | 99.00 | 0.10 | 33.97 |
| GG | $pSEq$ | 21.38 | 12.85 | 27.39 | **20.54** |
| | Stably Matched (%) | 97.30 | 98.10 | 98.00 | 97.80 |

Table 4: Average $SEq$ ($\downarrow$) and $Bal$ ($\downarrow$) at $N = 100$.

| $100 \times 100$, UU | $SEq$ | $Bal$ |
|---|---|---|
| GS | 1259.39 | 1709.53 |
| PolyMin | 153.35 | 952.85 |
| DACC | 194.65 | 988.02 |
| Power Balance | **49.41** | **909.73** |
| WN-80f/b+Hungarian | | |
| $pSEq/pBal$ | 257.99 | 1145.36 |
| $SEq/Bal$ | 68.36 | 919.75 |
| Stably Matched (%) | 89.4 | 80.8 |

Table 3 shows the results. Remarkably, there is a contrast between the model trained with DD and the others. The model with DD could hardly satisfy the one-to-one stable matching constraint when tested on UU/GG, and resulted in poor $pSEq$ scores. In contrast, the model with GG achieved satisfying $pSEq$ scores on UU/DD. Since GG generates preference lists based on a common preference score ($i/N$ for $i$-th agent) with noise, agents in GG tend to have similar preference lists (i.e., hard to assign optimally). A model trained with such hard samples works well even for the test samples drawn from other distribution. UU has also performed well owing to its non-biased sampling strategy. On the other hand, DD worst performed due to its highly biased generation strategy. From these results, we confirmed that WeaveNet has certain robustness in the distribution shift as long as training samples are competitive enough.

### 5.3    Demonstration with $N = 100$

We further demonstrate the capability of WeaveNet under a larger size of problem instances, $N = 100$. In this case, we found that WN-80f and WN-80b failed to yield one-to-one matchings for 13.4% and 19.8%, respectively (see the Table 9 in B.2 for details). To compensate for this problem, we applied the Hungarian algorithm (Kuhn, 1955) to surely binarize $\hat{m}$ into a one-to-one matching. Table 4 shows WeaveNet's relatively good $SEq$ and $Bal$ scores. Even with the help of the Hungarian algorithm, they were strongly penalized in $pSEq$ and $pBal$ due to the poor stable matching rate. In other words, we can potentially fill the large gap by better constraining the output.

Since this work is just a pilot study toward a practical differential assignment solver, there is still a lot of space for improvement. The proposed test protocol with stable matching will facilitate it since we can freely adjust the difficulty of the problem to develop and enhance the methods continuously.

## 6    Conclusion

This paper proposed a novel differential assignment solver, *WeaveNet*, and an evaluation protocol on two *strongly NP-hard* variants of stable matching. In the experiments, we demonstrated the advantage of *set encoder* and the two-stream architecture of Weavenet against the other learning-based methods. These techniques also achieved a better performance than the state-of-the-art algorithmic method when $N = 20$ and a comparative performance when $N = 30$. Furthermore, the asymmetric variants, *split batch normalization* with the side-identifiable code, enabled the method to work even with the strongly biased dataset of UD. We also confirmed that the proposed method does not work at $N = 100$, which will be an immediate task for this new field of differential assignment solver. We hope that this work becomes a starting point to open a new vista for real-world assignment problems.

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
