# OpenReview forum: "WeaveNet for Approximating Assignment Problems"
_NeurIPS.cc/2021/Conference — NeurIPS 2021 Submitted_

### Official Review · Reviewer_EJfv · 2021-07-12

**Rating:** 5
**Confidence:** 3

**Summary:**

This paper proposes a learning-based algorithm to tackle assignment problems, which have traditionally used approximation algorithms to solve. The paper details WeaveNet, a novel neural network architecture which includes three techniques of set-encoder, split batch normalization and feature weaving layer, to efficiently solve the NP-hard combinatorial problem. Experiments show competitive results compared to other learning-based algorithms.

**Main Review:**

Overall, the paper is clear and details the technical contributions of WeaveNet’s architecture and capabilities in handling flexible inputs. Experiments demonstrate that WeaveNet is the strongest algorithm compared to other learning-based algorithm baselines. Experiments also show the empirical improvements of each novel component of WeaveNet (set-encoder, two-stream architecture) and why these changes are necessary. WeaveNet’s deterioration in performance (in % stable matches and fairness) compared to traditional algorithm baselines for $N=100$ indicates that there could be limited scalability as $N$ increases (Table 4).

My main concern is how asymmetry (the case where $N \neq M$) affects the fairness objectives and is not included in experimental results.  The fairness objectives (Eqn 1 and 2) seem intuitive for the case where $N = M$, but not as intuitive when inputs are not equal in size. If I am interpreting the metric correctly, for the case where $N >> M$ and preferences are uniformly generated, it is natural that all $b\in B$, where $|B| = M$ are matched because and only highly preferred $a\in A$ where $|A| = N$ are matched. As a result, there could be a natural "skew" in the metrics that arises from this asymmetry and not unfairness: $P(m; B)$ is much larger than $P(m; A)$. Related to this, the experiments only show results for the case when $N=M$. Is this a limitation of the architecture or optimization process? It would be helpful for the authors to address these two points.

 Specific questions:
-	It seems WeaveNet consistently underperforms on the UD synthetic distribution. Is there any intuition on why this is the case?
-	How long does the training procedure take to run for WeaveNet? How does this compare to the runtime of traditional algorithms?



**Time Spent Reviewing:**

3

---

> ### Author Response · Authors · 2021-08-10
> **Author response to Reviewer Ejfv**
>
> We appreciate the reviewer for their insightful comments.
> We hope the following response resolves their concerns.
>
> > *My main concern is how asymmetry (the case where $N\neq M$ ) affects the fairness objectives ... not as intuitive when inputs are not equal in size ...*
>
> As the reviewer has concerned, there can be many different *intuitive* definitions for fairness, depending on the situation. That is the motivation of the learning-based solver (L4-6).
> The definition of *Eqn 1 and 2* was from past operations research studies, and many strong algorithms are hand-crafted. We adopted them to have a solid evaluation with such baselines.
>
> > *there could be a natural "skew" in the metrics that arises from this asymmetry and not unfairness*
>
> In that case, we can define another fairness metric, for example, by weighting $P(m;A)$ and $P(m;B)$ differently. However, hand-crafted algorithm is often inflexible even in such a slight change in objective functions.
>
> > *Is this a limitation of the architecture or optimization process?*
>
> The setting of $N=M$ does **not derive from any limitations by the architecture and optimization process**. We have followed the setting of traditional hand-crafted algorithms. Such studies assume $N=M$ since we can usually convert a $N\times M$ problem instance into an equivalent $N\times N$ problem instance by inserting dummy agents. An agent has no partner when matched to the dummy. On stable matching, the dummy agents are generated as those worst preferred by the agents on the other side.
>
> We noticed that we failed to describe the reason. We will clarify this point.
>
> ## Specific Questions
> > *It seems WeaveNet consistently underperforms on the UD synthetic distribution. Is there any intuition on why this is the case?*
>
> With UD, GS performs competitively to PowerBalance. Note that GS is an algorithm that outputs the maximally biased solution. Hence, it is strongly expected that there are few fair solutions with UD. Under such a situation, the challenge of minimizing $\mathcal{L}_{f/b}$ only encourages violation of stability. This is the *intuition on why this is the case*.
> Automatic hyper-parameter tuning might be a solution to this problem.
>
> > *How long does the training procedure take to run for WeaveNet? How does this compare to the runtime of traditional algorithms?*
>
> Please see Table 10 in Appendix for the runtime of the training procedure. It takes 10 to 110 hours, depending on the problem size and the number of layers. Since traditional hand-crafted algorithms do not require any training procedure, we cannot compare them.
>
> However, it takes a long time to hand-craft a good algorithm. For example, PowerBalance (2020) has been developed 31 years after the problem definition (Gusfield and Irving, 1989) and DACC (2016) 27 years. In this sense, a few hundred hours of training are negligibly short compared to the cost to hand-craft a good approximation algorithm.
>
> ## Summary
> > *Experiments show competitive results compared to other learning-based algorithms.*
>
> We believe this is just a typo. As the reviewer stated in the main review by themselves, *WeaveNet is the strongest algorithm compared to other learning-based algorithm baselines.*

---

> > ### Comment · Reviewer_EJfv · 2021-09-01
> > **Rebuttal response**
> >
> > I thank the authors for taking their time to clarify certain points, especially those surrounding the fairness objectives. However, I am unconvinced that the results are significant enough given the lack of generalizability to other settings (for large graphs where N = 100) combined with the fact that experimental results were not very strong against hand-crafted algorithms. Therefore, I will keep my score the same at a weak reject.

---

### Official Review · Reviewer_PXDn · 2021-07-16

**Rating:** 5
**Confidence:** 5

**Summary:**

This paper provides a differential approach to solve matching problems. Since matching is a highly combinatorial problem, two main elements must be provided for approximating such an NP-hard problem: permutation invariance (PI) and variable-size inputs (VS). Both elements are borrowed from PointNet and Deep Sets as follows. In the first "weaving" layer,  the matching preferences of both agents (eg. men and women) are weaved so that for each row of men (women) preferences we stack the women (men) preferences. Each  men-women and women-men weaving is independently fed into a set encoder which basically relies on 1D-convolution + max-pooling. Convolution (dot product when we have 1D convolution) provides a universal space where all men-women (women-men) sample pairing will be projected so that a given test problem can be solved. Solving a problem means to map it to a solution code. In this regard, given the latent features (after L Weave-layers), the last convolution must be learnt before  softmaxing.

Expermiental results show that the chosen (two-streams) architecture outperforms the state-of-the-art alternatives, including non-differtial methods for N close to 30. However, the proposed method does not scale for N=100 (even with the help of the Hungarian method for constraining the output).

**Ethical Concerns:**

Not applicable.

**Limitations And Societal Impact:**

Not applicable.

**Main Review:**

Originality: Are the tasks or methods new? Is the work a novel combination of well-known techniques? (This can be valuable!) Is it clear how this work differs from previous contributions? Is related work adequately cited?

A bit original, what is more novel is the two-stream architecture. It is clearly the best choice for small problem sizes (N=3, 5, 7, 9) when many of the alternatives decay exponentially in performance. For N=20, 30 the proposed architecture is still competitive.

Quality: Is the submission technically sound? Are claims well supported (e.g., by theoretical analysis or experimental results)? Are the methods used appropriate? Is this a complete piece of work or work in progress? Are the authors careful and honest about evaluating both the strengths and weaknesses of their work?

Quality. Moderate. Although the complementary stuff contains ablation studies etc, this reviewer does not identify a explanatory approach in terms of showing, for instance, how the weaving leads to find a code after (in general L = 2N) layers. Ok, I know that each layer is doing part of the job in terms of permutation invariance, as it happens for instance in PointNet. However, the authors should stress what are the "critical points" (critical lists of choices in this case). The answer to this question is partially given in the sythetic experiments (distributional choices), but the authors are loosing a golden chance to dig more in the need of more or less layers attending to how difficult is the problem in a principled way.

Clarity: Is the submission clearly written? Is it well organized? (If not, please make constructive suggestions for improving its clarity.) Does it adequately inform the reader? (Note that a superbly written paper provides enough information for an expert reader to reproduce its results.)

Not so clear. For instance, no explanation is given about the VS inputs. Regarding the weaving process, they should remark that a row of an agent is mixed with those of the other agent in column form.

Significance: Are the results important? Are others (researchers or practitioners) likely to use the ideas or build on them? Does the submission address a difficult task in a better way than previous work? Does it advance the state of the art in a demonstrable way? Does it provide unique data, unique conclusions about existing data, or a unique theoretical or experimental approach?

Significance. They are relatively important insofar they tell the community that for moderate values of N, deep learning can deal with an NP-Hard problem. However, as it is well known that the approximation of this problem via  making it continuous suffer from plateaus in the cost function, no insight is given in terms of how good is the deep learning approach (what are its limits and, specially, why).

**Time Spent Reviewing:**

10 hours

---

> ### Author Response · Authors · 2021-08-10
> **Author response to Reviewer PXDn**
>
> We appreciate the reviewer for spending their precious time on our work.
> It seems that they raised some concerns on the *explanatory approach* of the method.
> We also find some ignored contributions of the study.
> We hope the following response helps them resolve the concerns.
>
> ## Originality
>
> > *For N=20, 30 the proposed architecture is still competitive.*
>
> Although the proposed method *is still competitive* to the hand-crafted algorithms, **the method has a clear advantage against such hand-crafted baselines**. Namely, we can flexibly modify the input/output formats as well as loss functions.
> Please consider it at evaluating our work.
>
> We have tested the proposed method with these algorithms since they are strong baselines on an assignment problem developed over decades for the popular stable matching problem. As stated in L47 and L84, it is a proper benchmark but **our motivation lies in developing a differentiable assignment solver, and it outperforms any other learning-based methods with a large margin.**
>
> ## Quality
>
> > *this reviewer does not identify a explanatory approach in terms of showing, for instance, how the weaving leads to find a code after (in general L = 2N) layers.*
>
> This is a good question. We hope A.3 and Fig. 8 in Appendix answer to this point.
> A.3 and Fig. 8 explain that any $ij$-th code refers to the entire agents at every two layer-stacking. Hence, every agent passes the message to each other at every two stacked layers. Hence, we think stacking layers is similar to increase the number of "outer iterations" of traditional hand-crafted methods. Namely, each outer iteration scans all the elements once in a specific way (hand-crafted inner process). Similarly, each stacked layer (jointly with its previous layer) enables the entire message passing, which is an alternative to the inner process.
>
> We will emphasize this in the manuscript.
>
> ## Clarity
>
> > *no explanation is given about the VS inputs.*
>
> L144-148 and Fig. 3 clearly describe the critical point of this question. All the components consisting of the set-encoder are size-independent.
>
> > *they should remark that a row of an agent is mixed with those of the other agent in column form.*
>
> This is what the cross-concatenation does and is clearly remarked in L142-144 and Fig. 3, with the mathematical formulation at L149-153.
> Since all the processes are done horizontally or vertically, the size-independent property of the entire network is ensured only by the set-encoder's size-independency.
> We will emphasize this point in the manuscript.
>
> ## Significance
>
> > *no insight is given in terms of how good is the deep learning approach (what are its limits and, specially, why).*
>
> Although an insight *in terms of how good is the deep learning approach* is a quite important question and we also have interest in it, please notice that the motivation of this study is not on it. **We have aimed to develop a differentiable assignment solver**, which is flexible and would contribute to solving assignment problems involved in many deep learning applications. We are happy if the reviewer evaluates this paper based on what we tried to achieve rather than what we did not aim to do. Please also refer to the next response, which also relates to this comment.
>
> > *it is well known that the approximation of this problem via making it continuous suffer from plateaus in the cost function*
>
> It is helpful if the reviewer could provide a specific reference that explains such a property on the approximation of the fair stable matching problems.
> We also think it is a general open problem how a deep learning model overcomes such problems of annoying local minima and reaches a new-optimal solution. There are a series of discussions on how a deep neural network avoids such a problem [1,2], which is not the target of this study.
>
> - [1] J. Frankle et al., "The Lottery Ticket Hypothesis: Finding Sparse, Trainable Neural Networks", ICLR 2019 (the best paper)
> - [2] N. Liu et al., "Lottery Ticket Preserves Weight Correlation: Is It Desirable or Not?", ICML 2021

---

> > ### Comment · Reviewer_PXDn · 2021-08-28
> > **On significance**
> >
> > It seems to me that the authors have not investigated the drawbacks of the continuous matching solvers. One good example is the Gold & Rangaajan approach:
> >
> > S. Gold and A. Rangarajan, "A graduated assignment algorithm for graph matching," in IEEE Transactions on Pattern Analysis and Machine Intelligence, vol. 18, no. 4, pp. 377-388, April 1996, doi: 10.1109/34.491619.
> >
> > and the convergence proof:
> >
> > A Convergence Proof for the Softassign Quadratic Assignment Algorithm
> > Part of Advances in Neural Information Processing Systems 9 (NIPS 1996)
> >
> > In these papers it is clearly stated that the energy minimization landscape is subject to fake attractors, for instance a tendency to match notable nodes (vertices with many links). This is due to the quadratic nature of the cost function wrt the matching variables. Weave net has an interesting (max) non linearity but the point of how good is this non-linearity wrt to e.g. quadratic is not investigated. I think that this is very important since the quadratic flavour of the cost function is what smoothes the energy minimization landscape. Actually subsequent authors have pursued alternative non-linearities (log, kernerlized - see for instance "Graph matching and clustering using kernel attributes. Neurocomputing 113: 177-194 (2013)". This is a critical point not developed in the paper.

---

> > > ### Author Response · Authors · 2021-08-30
> > > **Response to the comment on significance**
> > >
> > > We appreciate the reviewer's active discussion for our paper and providing other exciting works.
> > > We understood that the first work by *Gold \& Rangarajan* proposes a method that solves the quadratic assignment problem based on energy minimization for each problem instance. The second paper (by Rangarajan and Yuille) provides theoretical proof on the convergence of such energy minimization under the doubly stochastic assumption (satisfied by repeating the Sinkhorn operations) on output. The third paper (by Lozano and Escolano) discusses the problem of plateaus, mentioned in the first review comment, on the problem of quadratic assignment. The paper proposes to use kernel functions to smooth the landscape of cost function. Note that we could not find the exact explanation of *fake attractors* in the above three papers, but the following reference [3] (cited in the third paper) seems to report it.
> > >
> > > - [3] Finch et al., "An Energy Function and Continuous Edit Process for Graph Matching", Neural Computation, 1998.
> > >
> > > The largest difference between WeaveNet and the algorithms discussed in the above papers is that WeaveNet minimizes the **expectation** of loss in training data. It does not apply "energy minimization for each problem instance" at inference time since we sometimes cannot access the complete input information and instance-wise optimization is not applicable (e.g., online assignment). Thus, the landscape of cost function matters only at training time and does not matter at inference (as if hand-crafted algorithms do not care about it). At training, the lottery ticket assumption explains how a deep learning method can optimize the model under such a complex landscape.
> > > We think that *the drawbacks of the continuous matching solvers* concerned by the reviewer only relate to the instance-wise optimization at inference, which is not the case of WeaveNet.
> > > Furthermore, **WeaveNet can collaborate with such instance-wise energy-minimization methods by providing good initial assignments**.
> > > In other words, the proposed backbone is not in competition with but beneficial to such methods.
> > >
> > > If purposely considering on instance-wise energy minimization, we can regard a neural network as a combination of a learnable kernel function with a linear classifier, similar to [4], which uses multilayer-perceptrons for a kernel function.
> > > Namely, with WeaveNet, the first $L-1$ layers learn the kernel function, and the last layer is the classifier. This means that the learned architecture can innately involve the kernel trick proposed in the third paper. In such context, meta-learning [5] can actively smooth the landscape of instance-wise energy minimization.
> > > Our focus is not on instance-wise energy minimization as stated above. However, it is compatible with such a rich deep-learning toolbox. Overall, a good backbone architecture, like WeaveNet, will contribute to any such extensions.
> > >
> > > - [4] Cho and Soul, "Kernel Methods for Deep Learning", NeurIPS2009
> > > - [5] Finn et al., "Model-Agnostic Meta-Learning for Fast Adaptation of Deep Networks", ICML2017
> > >
> > > Finally, as the reviewer recognizes *Weave net has an interesting (max) non linearity*, **how to convert such (max) non-linearity with a quadratic function is non-trivial**. WeaveNet automatically learns an algorithm that estimates a good solution for such a non-linear and non-quadratic optimization problem, which implies the significance of our work.

---

### Official Review · Reviewer_nn6T · 2021-07-16

**Rating:** 6
**Confidence:** 4

**Summary:**

The paper is about WeaveNet, a neural architecture, specialized to solve assignment problems that are NP-hard. Feature weaving layer, the key technical contribution by WeaveNet, is stacked to model frequent communication between elements for solving the combinatorial assignment problem. In particular, set encoder and the two-stream architecture of Weavenet achieve a better performance than the state-of-the-art algorithmic method.

**Ethical Concerns:**

None.

**Limitations And Societal Impact:**

Yes, the authors have seemed to adequately address the limitations and societal impact of their work.

**Main Review:**

The essence of the paper, hence the bottomline of what WeaveNet solves, is matchmaking. Simply put, imagine there are two groups, and members in each group want to be match-made to members from the other group. The matchmaking preference is described by an MxN matrix, and for two groups, the two matrices as an input are fed into WeaveNet. The input matrices are encoded and used to solve a combinatorial optimization problem.

The key contributions are:
1. Creation of new architecture for a neural net that solves combinatorial optimization problems (WeaveNet encodes two input matrices).
2. It is new that assignment problem, a type of combinatorial optimization problem, is solved by deep reinforcement learning.

Some technical comments are:
- The WeaveNet neural architecture is kind of new, and it solves really well when the degree of input complexity (say, N=20,30). For larger N, WeaveNet does not really generalize better performance gain for smaller N.

- Assignment problem is not solved by neural combinatorial optimization. Applying deep RL for new combinatorial optimization problems is a great attempt. The problem solved by WeaveNet, however, is limited.

- It is formulaic that the two input matrices are always encoded. For more general problems that require one matrix or three matrices as an input, WeaveNet is applicable to a narrow limited class of problems.

**Time Spent Reviewing:**

2

---

> ### Author Response · Authors · 2021-08-10
> **Author response to Reviewer nn6T**
>
> We appreciate the reviewer's insightful comments.
> We are pleased to see the positive feedback that recognizes the advantage of the learning-based assignment solver. On the other hand, there is a critical misunderstanding caused by our inappropriate presentation, and it can harm the reliability of the review. Namely, we do not use reinforcement learning (RL) and the model (including loss functions) is fully differentiable.
>
> We hope the following response resolves the misunderstanding and helps the reviewer re-evaluate our work.
>
>
> ## The key contributions
>
> > "It is new that assignment problem, a type of combinatorial optimization problem, is solved by deep reinforcement learning."
>
> We did not use reinforcement learning (RL). RL was used in (Gibbons 2019) (see L68). Instead, we have trained the model with an unsupervised fully-differentiable procedure (see L174) on the basis of (Li 2019)'s implementation. Hence, the major contribution lies in the network architecture, which we integrated multiple novel techniques of set-encoder, two-stream architecture, and split batch normalizetion.
>
> From the comment by the reviewer, we noticed that the keyword "end-to-end" is often used to connect a non-differentiable module by RL and is confusing. We will revise the manuscript at this point.
>
> The use of the Hungarian method in 5.3 may also cause this misunderstanding.
> There, we put the algorithm only at test time and not trained the model with it.
> As misunderstood by the reviewer, it is natural to consider the use of RL with our model; it will give a theoretical guarantee on the output (e.g., to make it one-to-one matching or stable matching with any non-differentiable post-processing). Our focus was on the ability of architecture itself, and we have eliminated such a post-process from the training procedure not to cover the weakness of the architecture unknowingly.
>
> ## Technical comments
>
> > *For more general problems that require one matrix or three matrices as an input, WeaveNet is applicable to a narrow limited class of problems.*
>
> This is a very interesting point. First of all, **WeaveNet can accept a variety of structured data with a small modification.**
>
> For example, it can process a single matrix as follows (without any modification in this case).
> Let $E$ be a $|V|\times |V|$ matrix, representing a set of directed edges in a graph $G(V, E)$. $E^\top$ is a transpose of $E$, representing a backward of the directed edges. Then, we can feed $E$ and $E^\top$ to the two-stream WeaveNet architecture. With the symmetric variant, the set-encoder processes the pair-wise relations of nodes regardless of the direction (forward and backward). Hence, it is suitable for an undirected graph. The asymmetric variant (L162) can differentiate the direction and is suitable for directed graphs.
> (Note that, $E$ is symmetric when $G$ is an undirected graph. Then $Z^A\_\ell=(Z^B\_\ell)^\top$ for any $\ell$ and we can reduce the computational cost).
>
> To feed more than three matrices (e.g., for $K$-partite matching), we can extend WeaveNet, for example, to a $K$-stream network, where cross-concatenation is also extended to cycle-concatenation (namely, concatenate $(Z^1\_\ell,Z^2\_\ell,\ldots,Z^K\_\ell)$ for the 1st stream, $(Z^2\_\ell,\ldots,Z^K\_\ell,~Z^1\_\ell)$ for the 2nd stream, ... and $(Z^K\_\ell,Z^1\_\ell,\ldots,Z^{K-1}\_\ell)$ for the $K$-th stream). The set-encoder can process each concatenated feature as the standard WeaveNet.
>
> Overall, the architecture of **WeaveNet is not formulaic but applicable for general structured data** in nature. We hope the reviewer find a further value from this response.

---

### Official Review · Reviewer_wb66 · 2021-07-16

**Rating:** 4
**Confidence:** 4

**Summary:**

The authors propose a learning-based fully differentiable architecture for assignment problems. The core module *feature weaving layer* consists of a *cross-concatenation* module and a *set-encoder*, modeling communication between the nodes. It can be stacked multiple times through the use of skip-connections, resulting in a deep network that is able to handle varying problem-sizes as input. The architecture is tested on two versions of the stable matching problem and compared against both learning-based and algorithmic baselines.


**Limitations And Societal Impact:**

Limitations as described in main review. No societal impact.

**Main Review:**



## Clarity
The paper has some minor issues regarding the presentation. While the visualizations are very helpful, I have the following suggestions for improvement:

- Line 94: In $m_{vw}=0$, $m$ is used as a matrix, but in the equation following line 104 it is used as a set ($\{a_i,b_j\}\in m$).

- line 96 \& 97:  Using dotted and dashed lines instead of colors would resolve problems with printing in grayscale.

- Line 217 \& 2018: Potential typo: "train" instead of "validation .

- Equation 7: Potential typo: start index of the sum is $1$ instead of $0$.

- Section 5.1: I was unable to find the specification of the dataset in this experiment.



## Originality
The architecture design incorporates some novel ideas such as the *cross-concatenation* and *split batch normalization*, while also using known techniques (general structure of the architecture is taken from DBM, the *set-encoder* is "inspired by DeepSet and PointNet").

## Quality
I have multiple concerns about the quality of this paper:

- Fairness with respect to baselines:

  - Line 70: "the implementation details are not completely explained". As DBM serves as a baseline later on, is this a reimplementation of DBM based on incomplete information? What exactly is missing?

   - Line 505: Only using two layers in a graph neural network massively limits its expressivity (oversmoothing indeed can be a problem, however, there exist various new techniques to reduce this problem). An ablation using more layers for the GNN would be useful in any case.

    - Line 470: Why is there a procedure for first fixing $\lambda_m$? The hyperparameters $\lambda_{m/s/f/b}$ only have meaning up to an arbitrary scale depending on the learning rate, so why not just immediately start out with $\lambda_m=1$ and then finetune $\lambda_{s/f/b}$?

- Presentation of results:

    - Figure 5, 6 \& 7:
        - The scale in this figure (subtracting the ideal values) is misleading in my opinion, overstating the significance of the improvement. E.g. in figure 7 for an agent size of $9$x$9$ the $0.25$ performance gain is not a "large margin" [line 52] when compared to the ideal value of $18.706$. I would suggest not subtracting the ideal value and instead reporting the ideal values as an optimal line in the plot.
        - Why is there a comparison to DBM-18 in Figures 6 \& 7? It is not reported in figure 5. I would suggest reporting all methods from figure 6 \& 7 also in figure 5 (including the XXX-18f/b), as the success rate of stable matchings is crucial in judging the meaningfulness of the pseudo-scores in figure 6 \& 7.

    - Table 10: Reporting the runtimes for the solvers as well would be important for judging the runtime o the presented architecture.

    - Random restarts: Errorbars are only reported for the hyperparameter-tuning in Fig 10-13. This would be more important for the main results (at least for the smaller problem instances), in which no statistics are reported.

## Significance
I do not think that the presented work in its current stage is very significant. While the idea of fully differentiable solvers is interesting especially in the case when the solver is part of a larger architecture (e.g. for feature extraction before the solver), this is not the focus of this paper. Instead, the main focus is training the architecture to improve upon the existing approximate solvers. While the results show that the architecture does not perform worse than the baseline solvers, by themselves they do not provide strong enough evidence that this architecture would perform much better than an algorithmic solver in a new setting, as in many of the experiments, the improvement upon the SOTA solver Power Balance is non-existing or very small.

While I believe that this is an interesting research direction, this paper does not meet the high standards of NeurIPS.
As a suggestion for future research, I believe that employing this architecture in a larger end-to-end trainable pipeline could make full use of the differentiability in order to produce significant results. One potential application for this could be multi-object tracking from image input.

**Time Spent Reviewing:**

10

---

> ### Author Response · Authors · 2021-08-10
> **Author response to Reviewer wb66**
>
> We appreciate the reviewer for spending their precious time on our work.
> It seems that their primary concern is the absence of *a larger end-to-end architecture* that will experimentally demonstrate the advantage of a learning-based model in flexibility against algorithmic methods. We hope our response clarifies the focus of this fundamental study and helps the reviewer re-evaluate our work.
>
> ## Significance
>
> > *While the idea of fully differentiable solvers is interesting especially in the case when the solver is part of a larger architecture...*
>
> It is controversial what kind of study is significant and what is not.
> Please notice that **this is a fundamental study rather than an applied study**.
> A fundamental study is significant (1) if the results give grounds of expectation for a wide variety of applications and (2) when the proposal's effect is evaluated systematically.
>
> For (1), as the reviewer expects an evaluation with a larger model, our study gives such an expectation.
> **As the first GAN paper[1] was tested only on MNIST, TFD, and CIFAR-10, many studies have been accepted to past first-tier ML conferences with the limited size of experiments [2,3,4,5]**. Hence, we believe that the absence of such a demonstration does not harm the significance immediately.
>
> - [1] I. J. Goodfellow, "Generative Adversarial Nets", NeurIPS2014
> - [2] M. Long, "Learning transferable features with deep adaptation networks", ICML2015
> - [3] C. Louizos et al., "The Variational Fair Autoencoder", ICLR2016
> - [4] D. Li et al., "Learning to Generalize: Meta-Learning for Domain Generalization", AAAI2018
> - [5] E Tzeng et al., "Adversarial Discriminative Domain Adaptation", CVPR2017
>
> For (2), we have established the systematic protocol based on NP-hard assignment problems with strong baselines. This is also a contribution of this work (L47). The protocol assesses the network's discriminative ability on higher-order combinatorial relationships among elements.
>
> Note that, although this is not the *larger architecture* experiment, we have provided the LibimSeTi distribution obtained from a real application (L214).
> As MNIST and CIFAR-10 have played an essential role in the early stage of deep learning research, **a solid evaluation protocol is vital for accumulating research in the future and should not be undervalued**.
>
> Note also that, in our experiments the total number of input patterns is $4.3\times 10^{16}$ even for $N=5$ cases (L245-248) and $3.73\times 10^{1880}$ for $N=30$, which is astronomically larger than 1024x1024 color images ($1.76\times 10^{13}$ patterns). Hence, **the problem is large enough from the viewpoint of machine learning**.
>
>
> > *they do not provide strong enough evidence that this architecture would perform much better than an algorithmic solver in a new setting*
>
> It is difficult to prepare *"a new setting"* in a true sense. Namely, when the setting is really new, a comparison with existing solvers is unfair, but we have no other reliable baselines. Then, we can only bring a demonstration, which cannot be objective evidence. Instead, we compared the method with two different variants of assignment problems and achieved the comparative result. The proposed architecture is designed for general assignment problems and not specialized to stable matching (other than loss functions). In this sense, the provided experiments are simulations of *a new setting* for our method and are already strong evidence.
>
> Note that a response to Reviewer nn6T would support that the stable matching problems are *a new setting* for the proposed architecture (see *Technical comments*). As discussed there, we can feed general graphs to WeaveNet in nature. In other words, the architecture is not specially designed to deal with stable matching problems. Hence, the model obtained the performance only by the loss function design. We will revise the manuscript to emphasize this point.
>
>
> ## Clarity
>
> We appreciate the reviewer's detailed comments.
> We will fix these issues. Note that some of them are not the cases, for which we give the response as follows:
>
> > *Line 217 \& 2018: Potential typo: "train" instead of "validation .*
>
> This is as stated and not a typo. We have described training data in the next paragraph (at L219), where 200k training iterations with batch size 8 yields 1.6M random sample generation for the training data. Note that all these samples hardly overlap since the total possible input patterns are $4.3\times 10^{16}$ even when $N=5$ (see L245-248).
>
> > * Section 5.1: I was unable to find the specification of the dataset in this experiment.*
>
> The detail of the dataset (the distributions) appears at L208-223, and the specification is quite clear. We also have submitted the code to generate these datasets with the random seed, which will be publicly available and reproducible.
>
>
> ## Originality
>
> > *The architecture design incorporates some novel ideas ..., while also using known techniques (general structure of the architecture is taken from DBM, the set-encoder is "inspired by DeepSet and PointNet").*
>
> It is also often controversial what is novel and what is not. However, any study is standing on the shoulders of giants, and  "*using known techniques*" **cannot be a flaw.**
> The novelty lies in the combination of such techniques. In our case, it is novel to use the set-encoder to the set of adjacency edges, where neither DBM nor any other graph-based neural network architectures do so. Stacking the set-encoder is also quite different from its original usage.
>
> ## Quality
> > *Line 70: ... As DBM serves as a baseline later on, is this a reimplementation of DBM based on incomplete information? What exactly is missing?*
>
> We have re-implemented DBM based on their incomplete information. We will emphasize this point in a future update.
> The missing information *exactly* was the number of channels for each layer, the order of activation and batch normalization, and the location and frequency of residual connections. Note that the original paper changes its hyper-parameter (e.g., the number of layers and initial learning rate) for each variant of the WTA problem (for each instance size and the local structure type). Hence, it is not clear what is the best architecture for DBM.
> In our study, all the architectures were regularized based on the parameter size for a fair comparison (see Table 6 in Appendix).
>
> > *Line 505: Only using two layers in a graph neural network massively limits its expressivity (oversmoothing indeed can be a problem, however, there exist various new techniques to reduce this problem). An ablation using more layers for the GNN would be useful in any case.*
>
> "*massively limits its expressivity*" is **not the flaw of the experimental setting, but the limitation of GNN**. The number of layers for GIN is decided by the original paper as the best setting for node classification (see L78-79).  (Oono 2020) has theoretically explained how the general GNN models (including GIN) lose their node-discriminative power, as the reviewer recognize.
> **Any specific reference should be provided** to claim that *"there exist various new techniques to reduce this problem"*.
>
> Note also that **we have had a preliminary experiment with a deeper model, but it did not work at all** as well as the two-layer model. The preliminary experiment was not designed for the presentation (we had not yet decided how to bring a fair comparison at that time). We will re-run it and add it.
>
> > *Line 470 : ... why not just immediately start out with $\lambda\_m$ and then finetune $\lambda\_{s/f/b}$?*
>
> It was necessary since we first fixed the entire learning rate to 0.0001 (L222),
>
> ### Figures 5, 6 \& 7
>
> > *... overstating the significance ...*
>
> As the footnote shows absolute values, we paid an effort to avoid such an overstatement. It was valid as supported by the fact that the reviewer could correctly understand the data.
> Since the ideal value of Bal ranges from 2.4 to 18.8, a plot with absolute values makes the difference from the ideal value invisible even for the worst method. The purpose of the visualization is only to show the trends changing along $N$.
>
> When we see the performance in a percentage, it looks a tiny difference. However, the trend actually results in a large margin at $N = 20$ and $N=30$ (see SSWN and WN(underlined) in Tables 7 and 8 in the Appendix).
> We will modify the explanation at this point.
>
> > *I would suggest reporting all methods from figure 6 \& 7 also in figure 5*
>
> The performance of DBM-18 at Fig. 5 was [1.00, 1.00, 1.00, 0.99] for $N=3,5,7,9)$ and mostly overlaps with WN-18 ([1.00, 1.00, 1.00, 1.00]). It is the same for other XXX-18 methods and the lines will be overlap. Nonetheless, we will clarify this in a future update.
>
> > *Table 10: Reporting the runtimes for the solvers as well ...*
>
> We will add such a comparison. Note that the running time data were shown as information for reproduction.
> The WeaveNet implementation (with python) is designed for readability and customizability, but not for running time.
> Nevertheless, a comparison in running time will be interesting for many readers.
>
> Honestly, WeaveNet runs 100 times slower than the hand-crafted algorithms that run on CPU with a full optimization (by Java). We will add such comparison and discussion to Appendix.
>
>
> > *Random restarts*
>
> As stated in L405-406, the preliminary experiment "demonstrated the stable behavior" and "For the other part, we have multiple settings, and we observed a stable trend in the results." Hence, the results are reliable, and they are not obtainable by cherry-picking or any other cheat.

---

### Decision · Program_Chairs · 2021-09-28

**Decision:**

Reject

**Comment:**

Even though the reviewers appreciated certain aspects of this paper, they raised multiple issues such as the scalability to larger graphs and misleading figures that were not resolved by the authors' response. Due to those issues, this paper does not meet the bar for acceptance at neurips.

**Consistency Experiment:**

NeurIPS has a long history of experimentation. In 2014, NeurIPS ran an experiment in which 10% of submissions were reviewed by two independent committees to quantify the randomness in the review process. This year, we repeated a variant of this experiment to see how the quality of the review process has changed over time.  This paper was part of the experiment and was therefore assigned to two committees (consisting of reviewers, an Area Chair, and a Senior Area Chair) that reached independent decisions.  If both committees made the same recommendation, this recommendation was followed. If a single committee recommended acceptance, the paper was accepted (with the exception of a few cases in which the other committee identified what we considered a fatal flaw, e.g., an error in a key result).

Both committees reached the same decision: **Reject**

The other committee assigned to the paper recommended **Reject**.  You can find the other set of reviews, along with any follow up discussion with the authors here:
https://openreview.net/forum?id=2rAiDBJgR_